# A Robust LiDAR SLAM Method for Underground Coal Mine Robot with Degenerated Scene Compensation

**Xin Yang** [1], **Xiaohu Lin** [1,2,*], **Wanqiang Yao** [1], **Hongwei Ma** [2], **Junliang Zheng** [1] and **Bolin Ma** [1]

[1] School of College of Surveying and Mapping Science and Technology, Xi'an University of Science and Technology, Xi'an 710054, China
[2] Shanxi Key Laboratory of Mine Electromechanical Equipment Intelligent Detection and Control, School of Mechanical Engineering, Xi'an University of Science and Technology, Xi'an 710054, China
[*] Correspondence: xhlin214@xust.edu.cn

**Abstract:** Simultaneous localization and mapping (SLAM) is the key technology for the automation of intelligent mining equipment and the digitization of the mining environment. However, the shotcrete surface and symmetrical roadway in underground coal mines make light detection and ranging (LiDAR) SLAM prone to degeneration, which leads to the failure of mobile robot localization and mapping. To address these issues, this paper proposes a robust LiDAR SLAM method which detects and compensates for the degenerated scenes by integrating LiDAR and inertial measurement unit (IMU) data. First, the disturbance model is used to detect the direction and degree of degeneration caused by insufficient line and plane feature constraints for obtaining the factor and vector of degeneration. Second, the degenerated state is divided into rotation and translation. The pose obtained by IMU pre-integration is projected to plane features and then used for local map matching to achieve two-step degenerated compensation. Finally, a globally consistent LiDAR SLAM is implemented based on sliding window factor graph optimization. The extensive experimental results show that the proposed method achieves better robustness than LeGO-LOAM and LIO-SAM. The absolute position root mean square error (RMSE) is only 0.161 m, which provides an important reference for underground autonomous localization and navigation in intelligent mining and safety inspection.

**Keywords:** underground coal mine robot; degenerated scene; LiDAR SLAM; intelligent mining

## 1. Introduction

Autonomous localization and navigation in an unknown underground environment are important and well-researched yet still active fields for mobile robots and are the key to the automation of intelligent mining equipment. With the rise of big data, cloud computing and artificial intelligence, intellectualization, as a disruptive innovative technology, has become the core driving force for the reform of basic industries around the world. Using intelligent mining equipment to drive the transformation and upgrading of the traditional mining industry can essentially enhance the core competitiveness of mining enterprises and promote the development of the traditional mining industry towards the goal of high efficiency, safety and sustainability [1].

Intelligent mining is the core technology to achieve high-quality development of the coal industry, which requires fast, accurate, automatic and full coverage of spatial data acquisition [2]. However, the traditional single-point measurement and stand-by three-dimensional (3D) laser scanning [3] data acquisition methods are labor-intensive, low in efficiency and time-consuming, which cannot meet the requirements of autonomous operation of a robot under complex conditions in coal mines. A mining robot based on SLAM technology can accurately and quickly construct a 3D map of an underground coal mine, and the map provides flexible and reliable assistance to coal mining robots for intelligent navigation and obstacle avoidance, which can be applied to work in the hazardous area under coal mines, automatic patrol inspection, remote dispatching, etc.

To speed up the development of underground space exploration technology, the Defense Advanced Research Projects Agency (DARPA) launched the Underground Space Challenge in 2018 [4,5], which aims to revolutionize the level of rescue and exploration in underground space. However, the high dust level, weak texture and low illuminance in the underground environment lead to unstable feature point extraction in visual SLAM. LiDAR SLAM, on the other hand, has good accuracy and robustness. However, it suffers from motion distortion and point cloud degeneration, being easy to degenerate in the narrow mining environment. The defects of LiDAR SLAM can be compensated for by fusing IMU information, which is not affected by structural features and great changes in the environment and can provide high-precision pose estimation in a short time. However, an IMU faces error accumulation over time. To compensate for the defects of a single sensor, multi-sensor fusion methods are increasingly being used to improve the robustness of state estimation in complex environments [6–8].

Recently, with the continuous improvement of autonomous driving solutions and the implementation of the Xihe Plan (Seamless Navigation and Positioning Service in All Space, All Time, Indoor and Outdoor) of the Beidou Project, the demand for intelligent sensing is increasing [9–12]. Extensive studies have investigated the autonomous navigation and map building of robots, which can be divided into loose-coupled based methods and tight-coupled based methods. (1) The loose-coupled based method estimates pose by LiDAR and IMU data, which is efficient, but the error feedback mechanism is not fully utilized. Representative works are LOAM, LeGO-LOAM, Cartographer, etc. LOAM [13] is a pioneering framework for real-time odometry and mapping by applying point-to-line and point-to-plane registration. However, it uses only LiDAR odometry, which has no back-end optimization and loopback detection. LeGO-LOAM [14] performs clustering and segmentation on plane points to extract ground features and applies two-step optimization for pose estimation, which improves the operation efficiency on the lightweight platform. However, IMU data are only used to provide prior information for scan matching and point cloud distortion removal, which is not used as an observation constraint for optimization. Cartographer [15] uses the Unscented Kalman Filter (UKF) algorithm to fuse multi-source data for pose estimation. However, the back-end optimization is still constrained by point cloud matching, and it is easy to fail in the degenerated scene or when LiDAR rotates rapidly. (2) The tight-coupled based method fuses LiDAR and IMU observations together and then estimates the pose of the mobile robot. Its advantage is that the observation data of LiDAR and the IMU are fully utilized to reduce the cumulative error. However, it increases the amount of calculation, and the failure of one sensor may lead to the failure of the whole system. Koide et al. [16] proposed a real-time 3D mapping framework based on global matching cost minimization using LiDAR-IMU tight coupling, which achieves accurate and robust localization and mapping in challenging environments. FAST-LIO [17] is a proposed efficient and robust LIO framework based on tightly coupled iterative Kalman filters. However, the system discards the influence of historical data on the current state and global pose correction cannot be performed. LIOM [18] is a tight-coupled LiDAR/IMU localization and mapping framework based on graph optimization. To achieve real-time and consistent estimation, the moving sliding window method is used to marginalize the old pose. Meanwhile, the preintegration information of an IMU is used to eliminate the motion distortion of the point cloud and jointly minimize the error function of LiDAR and IMU measurement. However, this method still takes time to construct constraints and batch optimization in local windows. LINS [19] is an improvement on LIOM and proposes a tight-coupled LiDAR/IMU localization and mapping method based on an iterative error state Kalman filter algorithm, which overcomes the loss of accuracy caused by discarding high-order errors in the linearization process of an extended Kalman filter. Compared with the LIOM algorithm, LINS has improved operation efficiency. However, the accuracy of LINS depends on the structured environment. To improve accuracy and time efficiency, LiLi-OM [20] is a proposed tight-coupled LiDAR/IMU SLAM method using keyframe-based sliding window optimization, which can support solid-state LiDAR

and mechanical LiDAR. However, point cloud matching between adjacent frames degenerates in symmetrical tunnels and corners. LIO-SAM [21] is a proposed tight-coupled LiDAR/IMU fusion framework via smoothing and mapping. However, the IMU data are only used for joint optimization without considering the impact of environmental degeneration on SLAM results. Kim et al. [22] developed an autonomous driving robot for underground mines using IMUs, LiDAR and encoders. It fuses three types of sensors and achieves high accuracy in estimating the location of autonomous robots in underground mines. Miller et al. [23] used multiple quadrupedal robots to explore and map the inside of a mine tunnel. They demonstrated the feasibility and functionality of the method in laboratory and field tests. Mascarich et al. [24] combined multiple sensors to develop an autonomous driving robot that performs autonomous driving tasks such as exploring the tunnel mapping environment. Since LiDAR SLAM uses the geometry structure of the environment to perform localization and mapping, it is vulnerable to geometrically degenerated environments such as open space [25] and long, straight tunnels [26,27], especially when lacking enough constraints in the shotcrete surface and symmetrical roadway in an underground coal mine. Extensive studies have investigated the degeneration problems of SLAM—f or example, eliminating the degeneration by restricting movement [28,29], minimizing spatial degeneration components in the direction of degeneration [30], adding a plane feature constraint [31–35], sensor fusion [36–41], etc. Zhang et al. [42] integrated IMU and odometry information into the Cartographer's mapping process, which improved the robustness of the algorithm in a long corridor environment. The online method proposed in [43] explained the impact of environmental degeneration on pose estimation. However, this method only uses the non-degenerate direction solution and does not deal with the degenerate direction state.

Although great efforts have been devoted to achieving a robust LiDAR SLAM method for underground coal mines, the shotcrete surface and symmetrical roadway in underground coal mines make LiDAR SLAM prone to degeneration, which leads to the failure of mobile robot localization and mapping. Therefore, a robust LiDAR SLAM method is proposed for an underground coal mine robot with degenerated scene compensation, which detects and compensates for the degenerated scenes by integrating LiDAR and IMU data. Extensive experiments with qualitative and quantitative analyses have verified the accuracy and efficiency of the proposed method, which can realize accurate and reliable SLAM in underground coal mines and provide a theoretical reference and technical support for intelligent mining and safety inspection in coal mines. The main contributions of this paper are as follows:

1.  The unknown linear equation is added to the state optimization equation as the disturbance model to detect the direction and degree of degeneration caused by insufficient line and plane feature constraints.
2.  The IMU pose is used to compensate for ill-conditioned components in the direction of degeneration, which cannot be determined directly by scan matching. LiDAR rotation state degeneration is compensated for by projecting IMU poses onto plane features. When degeneration is also detected in the translation direction, the compensated rotation state and IMU translation state are fused into a new LIDAR pose, which is then used for scan-to-submap matching to achieve two-step degeneration compensation.
3.  A tightly coupled LiDAR/IMU fusion framework is implemented based on factor graph optimization. The IMU measurements and LiDAR point cloud features are jointly optimized in a sliding window, which improves the accuracy and robustness of SLAM in the underground coal mines with the shotcrete surface and symmetric roadway environment.

## 2. Materials and Methods

### 2.1. System Configuration

In this paper, a data acquisition platform is designed based on an underground coal mine mobile robot equipped with LiDAR and an IMU, as shown in Figure 1. The proposed

method defines the mobile robot coordinate system *B* as a reference, which is consistent with the LiDAR coordinate system *L*. The IMU coordinate system is defined as *I*. The world coordinate system is defined as *W*. The origin of the world coordinate system was set to the center of the mobile robot at the time of system initialization, and the Z-axis is opposite to the gravity direction in the world coordinate system. Table 1 shows the detailed specifications of the mobile robot system. An Autolabor Pro1 robot was used as the carrier to carry the VLP-16 LiDAR and Ellipse2-N IMU. External parameters between LiDAR and IMU ware calibrated in advance [44]. Allen variance was calculated to determine the degree of trust in the diagonal velocity and acceleration observations of the system. Time synchronization between the IMU and LiDAR was triggered by pulse per second (PPS). Then, the proposed method was implemented by C++ based on the robot operating system (ROS), and the nonlinear optimization was implemented using the Ceres library. The computer used in the experiment was AMD Ryzen3 3200G with DDR4 8GB.

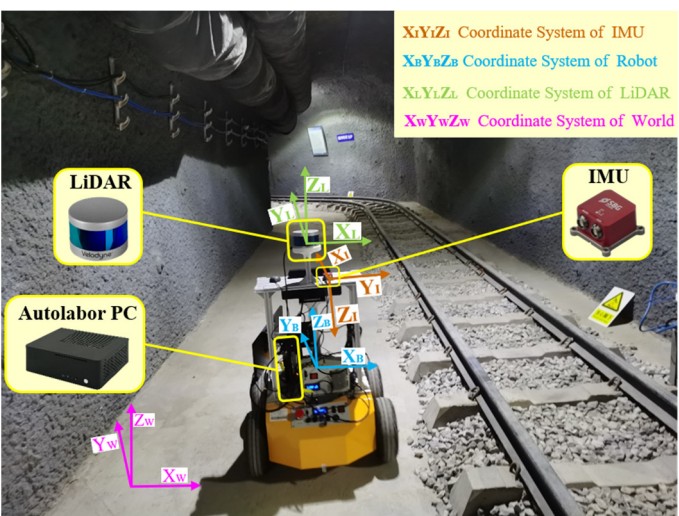

**Figure 1.** Data acquisition platform and environment.

**Table 1.** Specifications of data acquisition platform.

| Equipment | Type | Specifications |
|-----------|------|----------------|
| LiDAR | VLP-16 | Scanning frequency: 10 Hz<br>Operating range: 0.2~150 m |
| IMU | Ellipse2-N | Output frequency: 200 Hz<br>Error: Roll/Pitch $\pm 0.1°$, Yaw $\pm 0.5°$ |
| Controller | Autolabor PC | CPU: AMD Ryzen3 3200 G<br>Memory: DDR4 8 GB |
| Robot | Autolabor Pro1 | Driving mode:4WD Speed: 0.8 m/s<br>Applicable terrain: All terrain |

### 2.2. Method outline

The outline of the proposed method is shown in Figure 2, which can be divided into three parts. (1) Preprocessing: IMU pre-integration is used to remove point cloud distortion. The non-ground point cloud is clustered and segmented from the real-time collected point cloud, and then, the line and plane features are extracted. (2) Front-end: The pose from the IMU between consecutive frames of LiDAR is used to provide an initial value for the feature matching. The disturbance model is used to detect the direction and degree of degeneration caused by insufficient line and plane feature constraints. The IMU poses are projected onto the plane features to compensate for the LiDAR rotation state. The IMU translation state is fused with the compensated rotation state to form a new LiDAR pose

as the initial value for local map matching. (3) Back-end: The IMU pre-integration factor, LiDAR odometry factor and loop closure factor are used to construct the factor graph, and the new related variable nodes are optimized by the factor graph. The input of the proposed method is LiDAR point cloud and IMU data, and the output is the trajectory and map.

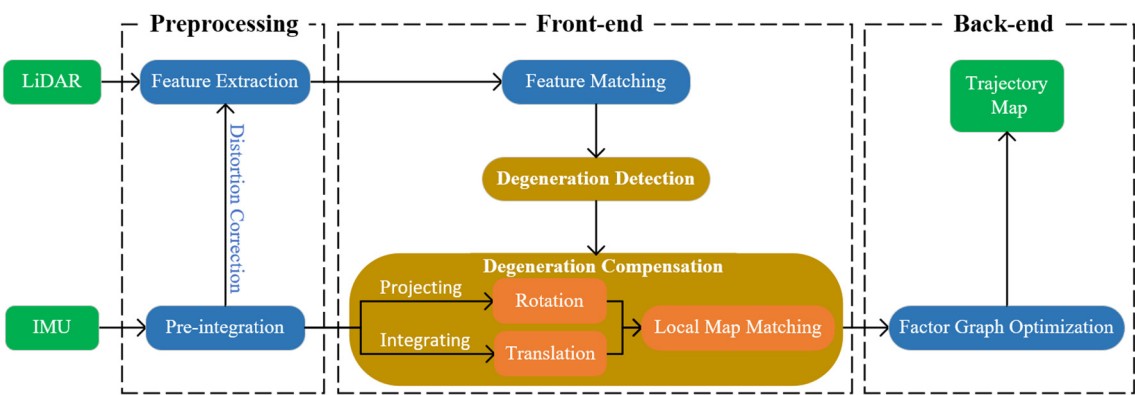

**Figure 2.** Schematic diagram of the proposed method.

### 2.3. Data Preprocessing

The IMU pre-integration method proposed in [18] was used to obtain the relative motion between LiDAR scan frames for distortion removal. In addition, IMU poses were also used to provide initial values for LiDAR odometry optimization [45]. The acceleration and angular velocity measured by the IMU can be expressed as $\widetilde{\omega}_{WB}^{B}(t_i^n)$ and $\widetilde{\alpha}^{B}(t_i^n)$, respectively, and the IMU measurement model is as follows:

$$\widetilde{\omega}_{WB}^{B}(t_i^n) = \omega_{WB}^{B}(t_i^n) + b_g(t_i^n) + \eta_g(t_i^n) \tag{1}$$

$$\widetilde{\alpha}^{B}(t_i^n) = R_{WB}^{T}(t_i^n)\left(\alpha^{W}(t_i^n) - g^{W}\right) + b_\alpha(t_i^n) + \eta_\alpha(t_i^n) \tag{2}$$

where $\omega_{WB}^{B}(t_i^n)$ is the instantaneous angular velocity of $I$ relative to $W$, $R_{WB}^{T}(t_i^n)$ is the rotation matrix from the world coordinate system to the IMU coordinate system, $\alpha^{W}(t_i^n)$ is the instantaneous acceleration in the world coordinate system, $b_g(t_i^n)$ and $b_\alpha(t_i^n)$ are the deviation of the gyroscope from the acceleration and $\eta_g(t_i^n)$ and $\eta_\alpha(t_i^n)$ are the random noise. According to the kinetic model of the IMU, the discrete integral method was used to integrate $\omega_{WB}^{B}(t_i^n)$ and $\alpha^{W}(t_i^n)$ in the IMU sampling interval $\Delta t$:

$$R_{WB}\left(t_i^{n+1}\right) = R_{WB}(t_i^n) \cdot \exp\left[\left(\left(\widetilde{\omega}_{WB}^{B}(t_i^n) - b_g(t_i^n) - \eta_g(t_i^n)\right) \cdot \Delta t\right)^{\wedge}\right] \tag{3}$$

$$v_{WB}\left(t_i^{n+1}\right) = \begin{aligned} &v_{WB}(t_i^n) + g^{W} \cdot \Delta t + \\ &R_{WB}(t_i^n)\left(\widetilde{\alpha}^{B}(t_i^n) - b_\alpha(t_i^n) - \eta_\alpha(t_i^n)\right) \cdot \Delta t \end{aligned} \tag{4}$$

$$p_{WB}\left(t_i^{n+1}\right) = \begin{aligned} &p_{WB}(t_i^n) + v_{WB}(t_i^n) \cdot \Delta t + \tfrac{1}{2}g^{W} \cdot \Delta t^2 + \\ &\tfrac{1}{2}R_{WB}(t_i^n)\left(\widetilde{\alpha}^{B}(t_i^n) - b_\alpha(t_i^n) - \eta_\alpha(t_i^n)\right) \cdot \Delta t^2 \end{aligned} \tag{5}$$

where " $\wedge$ " represents the transformation of a three-dimensional vector into an antisymmetric matrix. The re-parameterized results of IMU pre-integration between two keyframes prevent the repeated integration of IMU observations and improve the reliability of the algorithm.

The point cloud obtained by LiDAR in real-time inevitably has distortion due to the rotation mechanism of LiDAR and the nonlinear motion of the sensor. Matching with a distorted point cloud will result in a large drift error. The continuous IMU states $X_k^{W}$ and

$X^W_{k+1}$ closest to the current LiDAR point cloud timestamp were obtained by pre-integration, and then, the IMU state $X^W_{curr}$ at $t_{curr}$ in the world coordinate system was determined by the linear interpolation method:

$$X^W_{curr} = X^W_{k+1} \times \frac{t_{curr} - t_k}{t_{k+1} - t_k} + X^W_k \times \frac{t_{k+1} - t_{curr}}{t_{k+1} - t_k} \tag{6}$$

The motion distortion $\Delta p^{start}_{curr}$ of the current LiDAR cloud point was calculated in the start LiDAR point cloud coordinate system:

$$\Delta p^{start}_{curr} = T^L_W \left[ p^W_{curr} - \left( p^W_{start} + v^W_{start} \times t_{curr-start} \right) \right] \tag{7}$$

where $T^L_W$ is the transformation matrix from the LiDAR coordinate system to the world coordinate system and $t_{curr\_start}$ is time of laser line scanning in the frame. Then, the LiDAR point cloud could be converted to the start of the LiDAR point cloud coordinate system. The coordinates after the distortion removal can be calculated by $\Delta p^{start}_{curr}$.

In this paper, the features were described using the local roughness of the point cloud. The range image was used to calculate the roughness $c$ of point $S$ in its neighborhood in the same row, the non-ground points with larger roughness $c$ were marked as line feature points, and the points with smaller roughness $c$ were marked as plane feature points. The extraction results of line and plane feature points in the preparation roadway by the proposed method are shown in Figure 3. Many green plane feature points were extracted, but only a few blue line feature points were extracted in the underground coal mine environment. The coordinate system was the robot coordinate system, and the robot moved in the Y-axis direction. These arrows are the three-axis constraints of line and plane features in pose estimation.

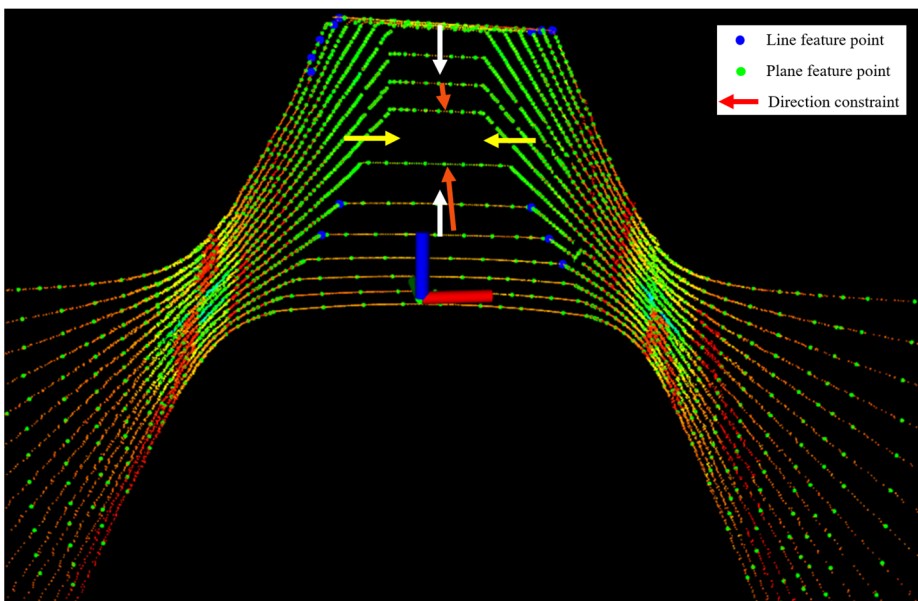

**Figure 3.** The extraction results of line and plane feature points in the preparation roadway. The red, white and yellow arrows are constraints in the three-axis direction.

*2.4. Front-End Odometry*

In this paper, we matched the point clouds $S_t$ and $S_{t+1}$ at moments $t$ and $t+1$ to estimate the pose $T^t_{t+1}$. Two matching points $\left( s^l_{t,i}, s^l_{t,j} \right)$ were found in the line feature points $L_t$ and three matching points $\left( s^p_{t,i}, s^p_{t,j}, s^p_{t,k} \right)$ were found in the plane feature points $P_t$. We established constraints on the minimum point-to-line and point-to-plane distances, respectively.

$$d_{L_{t+1,n}} = \frac{\left\| \left( s^l_{t+1,n} - s^l_{t,i} \right) \times \left( s^l_{t+1,n} - s^l_{t,j} \right) \right\|}{\left\| s^l_{t,i} - s^l_{t,j} \right\|} \tag{8}$$

$$d_{P_{t+1,n}} = \frac{\left\| \left( s^p_{t+1,n} - s^p_{t,i} \right) \cdot \left( s^p_{t,i} - s^p_{t,j} \right) \times \left( s^p_{t,i} - s^p_{t,k} \right) \right\|}{\left\| \left( s^p_{t,i} - s^p_{t,j} \right) \times \left( s^p_{t,i} - s^p_{t,k} \right) \right\|} \tag{9}$$

where $d_{L_{t+1,n}}$ and $d_{P_{t+1,n}}$ are the distances from the point-to-line and the point-to-plane, respectively. We obtained the function of $d$ to construct the objective function.

$$f\left( S^l_t, S^l_{t+1}, S^p_t, S^p_{t+1}, T^t_{t+1} \right) = \min \left\{ \sum d_{L_{t+1,n}} + \sum d_{P_{t+1,n}} \right\} \tag{10}$$

The pose $T^t_{t+1}$ cloud be solved by minimizing $f\left( S^l_t, S^l_{t+1}, S^p_t, S^p_{t+1}, T^t_{t+1} \right)$ through a nonlinear iterative objective function.

We used a two-step optimization for pose estimation. Point-to-line and point-to-plane scan matching were performed to find the transformation between two scans. Considering that plane features are more numerous than line features in the underground coal mine, the pose parameters $[\theta_{roll}, \theta_{pitch}, t_z]$ were preferentially estimated by matching many plane features. Then, the remaining parameters $[\theta_{yaw}, t_x, t_y]$ were estimated by line features while using $[\theta_{roll}, \theta_{pitch}, t_z]$ as constraints. The two-step pose estimation method optimizes the pose and effectively suppresses the pose drift. The initial value of iterative calculation was provided by IMU pre-integration between scan frames, effectively reducing the number of iterations. The Levenberg Marquardt method was used for pose optimization.

It is difficult for mobile robots to establish the data association of matching points in underground coal mines with insufficient structural features, resulting in insufficient direction constraints in the state space. The normal equation matrix of the point cloud matching solution is ill-conditioned. The degeneration factor was used to detect and judge the degeneration direction, and the linear optimization problem was constructed as follows.

$$\underset{x}{\arg\min} \| Ax - b \|^2 \tag{11}$$

The optimization problem corresponds to the overdetermined systems of linear equations $Ax = b$. The optimal solution was obtained by solving the overdetermined linear equations with the least-squares method. The unknown linear equation was added to the state optimization equation as the disturbance model of the degeneration problem to detect the direction and degree of degeneration. The degeneration factor $D$ was defined as a quantity only related to the eigenvalue $\lambda_i$ of $A^T A$, as shown in the following expression:

$$D = \lambda_{\min} + 1 \tag{12}$$

where $\lambda_{\min}$ is the smallest eigenvalue. Singular value decomposition (SVD) was used to obtain the six eigenvalues of $A^T A$, and their eigenvectors correspond to the $[\theta_{roll}, \theta_{pitch}, \theta_{yaw}, t_x, t_y, t_z]$. Feature matching fails when $D$ is less than the degeneration factor threshold $D_{thr}$, which can be obtained from the measured data of the actual environment. The statistical result of the degeneration factor in the indoor corridor environment is shown in Figure 4. We counted the degeneration factors of degeneration and non-degeneration scenes. The statistics present a clustering effect and can take the cut-off clustering value $D_{thr} = 180$ as the threshold.

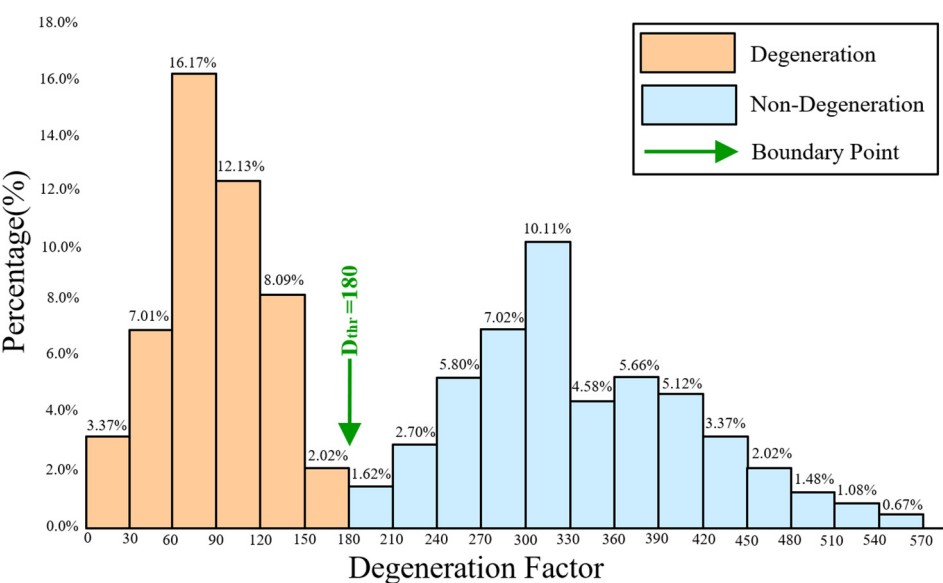

**Figure 4.** The statistical result of degeneration factor.

Figure 3 shows the constraints of line and plane feature points extracted from the scanning point cloud acquired in real time. A large number of plane feature points provide sufficient (red, white and yellow arrows) constraints on $[\theta_{roll}, \theta_{pitch}, t_z]$. Generally, these degrees of freedom do not suffer from degeneration problems. However, the extracted line feature points are few and the constraints provided for $[\theta_{yaw}, t_x, t_y]$ are insufficient, which is prone to degeneration. The sidewalls in the roadway are unsuitable to provide vertical constraints, and relying only on the lateral constraints of the LiDAR point cloud during turns can easily lead to incorrect pose estimation.

The scan matching ill-conditioned component caused by degeneration is compensated for by the pose from the IMU. The number of line feature points extracted from the LiDAR point cloud is far smaller than the plane feature points in the underground coal mine. Therefore, the proposed method uses the pose from the IMU to compensate for the degeneration rotation. The rotation state of the IMU pre-integration is accurate in a short time, but plane constraints may not be maintained. The rotation state with sufficient constraints is obtained by feature matching after degeneration compensation. Then, the compensated rotation state and the IMU translation state are fused into a new pose from LiDAR when the translation direction of the degeneration is detected. Finally, the LiDAR pose is used as the initial value of local map matching to pose optimization. It effectively reduces the drift of system pose estimation and improves the accuracy of localization and mapping.

In this paper, $q_{IMU}$ and $q_{LiDAR}$ represent the rotation state of the IMU and LiDAR, respectively; $q_{IMU}^*$ and $q_{LiDAR}^*$ are conjugates of $q_{IMU}$ and $q_{LiDAR}$; $\alpha$ represents the degenerated vector in the direction of rotation; and the normal vectors updated by the IMU poses and plane features are represented by $v_{IMU}$ and $v_{LiDAR}$, respectively, as shown in the following expression:

$$(0, v_{IMU}) = q_{IMU} \otimes (0, \alpha) \otimes q_{IMU}^* \tag{13}$$

$$(0, v_{LiDAR}) = q_{LiDAR} \otimes (0, \alpha) \otimes q_{LiDAR}^* \tag{14}$$

The rotation state of the current point cloud after degeneration compensation is represented by $q_L$, as shown in the following expression:

$$\begin{aligned} q_L &= q_{LiDAR} + \Delta q \\ &= q_{LiDAR} + \frac{(\|v_{IMU}\|\|v_{LiDAR}\| + v_{IMU} \cdot v_{LiDAR}, v_{IMU} \times v_{LiDAR})}{\|(\|v_{IMU}\|\|v_{LiDAR}\| + v_{IMU} \cdot v_{LiDAR}, v_{IMU} \times v_{LiDAR})\|} \end{aligned} \tag{15}$$

where $\Delta q$ is calculated by projecting the pose from IMU preintegration to the shortest arc length parallel to the plane feature. Then, $\Delta q$ is used to compensate for ill-conditioned vectors relative to the rotation direction that cannot be estimated by feature matching. For the degeneration in the translation direction, the IMU translation vectors combined with the compensated rotation vectors are used to reconstruct the point cloud state, which can be used as the initial value of the optimization iteration for local map matching. The pose $T_L$ of the current point cloud can be represented as follows:

$$
T_L = \begin{bmatrix}
1 - 2q_{L_y}^2 - 2q_{L_z}^2 & 2q_{L_x}q_{L_y} - 2q_{L_w}q_{L_z} & 2q_{L_x}q_{L_z} + 2q_{L_w}q_{L_y} & X_{L_x} \\
2q_{L_x}q_{L_y} + 2q_{L_w}q_{L_z} & 1 - 2q_{L_x}^2 - 2q_{L_z}^2 & 2q_{L_y}q_{L_z} - 2q_{L_w}q_{L_x} & X_{L_y} \\
2q_{L_x}q_{L_z} - 2q_{L_w}q_{L_y} & 2q_{L_y}q_{L_z} + 2q_{L_w}q_{L_x} & 1 - 2q_{L_x}^2 - 2q_{L_y}^2 & X_{L_z} \\
0 & 0 & 0 & 1
\end{bmatrix}
\tag{16}
$$

where $X_L$ represents the translation vector of the IMU, which can be converted into the translation of the current point cloud by the external parameters of LiDAR and the IMU. Local map matching adopts line and plane feature matching [46], which not only corrects the degeneration in the translation direction inter-frame matching but also limits the error accumulation of the LiDAR odometry.

The degeneration compensation algorithm proposed in this paper is shown in algorithm 1. The input values are the LiDAR pose $T_{LiDAR}$, the IMU pose $T_{IMU}$, the degeneration factor $D$ and the degeneration factor threshold $D_{thr}$. If the degeneration factor $D_r$ of the rotation direction is lower than the degeneration threshold $D_{thr}$, the rotation state in the IMU pose is extracted and projected to the plane features to compensate for the degeneration of the rotation direction. If the degeneration factor $D_t$ in the translation direction is lower than the degeneration threshold $D_{thr}$, the compensated degeneration rotation state $q'_{LiDAR}$ or the non-degeneration rotation state is fused with the IMU translation state to obtain a new LiDAR pose. Finally, the LiDAR pose is used as the initial value of local map matching to update the pose $T'_{LiDAR}$.

---

**Algorithm 1.** degeneration compensation

---

1:   **input**: $T_{LiDAR}$, $T_{IMU}$, $D$, $D_{thr}$;
2:   **output**: $T'_{LiDAR}$;
3:   **if** $D_r < D_{thr}$ **do**
4:       Compute $q_{LiDAR}$, $q_{IMU}$, $X_{LiDAR}$, $X_{IMU}$ of $T_{LiDAR}$, $T_{IMU}$;
5:        Compute $v_{LiDAR}$ and $v_{IMU}$ of $q_{LiDAR}$, $q_{IMU}$ based on (13) and (14);
6:       Compute $q'_{LiDAR}$ of $v_{LiDAR}$ and $v_{IMU}$ based on (15);
7:   **end**
8:   **if** $D_t < D_{thr}$ **do**
9:       Construct $T'_{LiDAR}$ from $q'_{LiDAR}$ and $X_{IMU}$ based on (16);
10:      Local map matching and updated $T'_{LiDAR}$;
11:      Return $T'_{LiDAR}$;
12:  **end**

---

### 2.5. Factor Graph Optimization

In this paper, factor graph optimization is used for data fusion, and the IMU pre-integration between consecutive keyframes is put into the factor graph. To reduce the redundant keyframes and improve the efficiency of the algorithm, a multiple keyframe selection strategy based on the Euclidean metric, rotation angle and overlap of the point cloud scan matching is introduced. The LiDAR odometry factor is constructed by the pose constraints between consecutive keyframes. The back-end optimization uses the GICP algorithm for matching. The loop closure factor is formed by the local loop closure and the global loop closure, which is introduced to construct the factor graph. Local loop closure is the intervisibility relationship between keyframes, which provides more constraints and improves the robustness of the system to LiDAR rotation motion when the mobile robot

turns. The global loop closure means that the mobile robot returns to the revisit position and establishes a loop closure constraint to reduce the global error accumulation. A new robot pose node is added to the factor graph when the change of robot pose exceeds the user-defined threshold. The factor graph is optimized upon the insertion of a new node using incremental smoothing and mapping (iSAM2) with the Bayes tree [47]. A schematic diagram of factor graph optimization by the proposed method is shown in Figure 5.

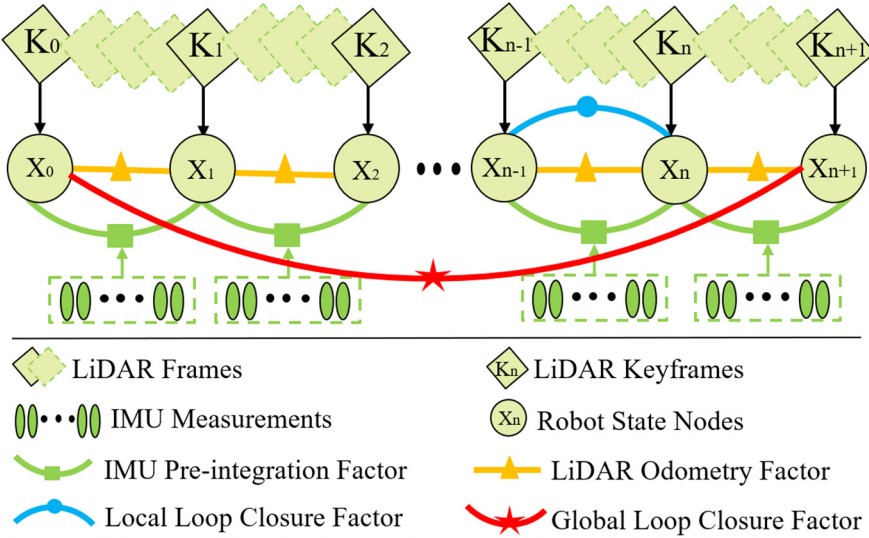

**Figure 5.** Schematic diagram of factor graph optimization by the proposed method.

## 3. Experimental Analysis

In order to verify the proposed method, qualitative and quantitative analysis experiments were undertaken with the data from an indoor corridor and an underground coal mine collected by a self-designed mobile robot platform and compared with other state-of-the-art LiDAR SLAM methods.

### 3.1. Qualitative Analysis

#### 3.1.1. Qualitative Analysis with Indoor Corridor

The mobile robot ran in the indoor corridor according to the path $A \rightarrow B \rightarrow C \rightarrow B \rightarrow A$, and the total length of the trajectory was 0.15 km, as shown in Figure 6a, which shows the localization and mapping results by the proposed method. It can be seen that the point cloud map constructed by the proposed method directly and accurately reflects the actual situation of the indoor corridor environment. The AB and BC segments are typical degenerate scenes with equal width on both sides, which can easily cause matching failure. The robot mistakenly believed it did not move, resulting in it mapping a length smaller than that of the actual trajectory. LIO-SAM drifted in the X-axis direction of the AB and BC segments, as shown in the yellow box in Figure 6b, and there is a ghosting in the mapping. Figure 6c shows the localization and mapping results by LeGO-LOAM. It can be seen that LeGO-LOAM has significantly degenerated in the Y-axis direction of the AB and BC segments, and the trajectory is shortened in the forward direction of the robot. There were matching errors in the pose estimation process, resulting in large localization drift and mapping ghosting. The proposed method detected and compensated for the degeneration in all directions. The trajectory was consistent with the actual motion of the mobile robot and the deviation of the point cloud map was small.

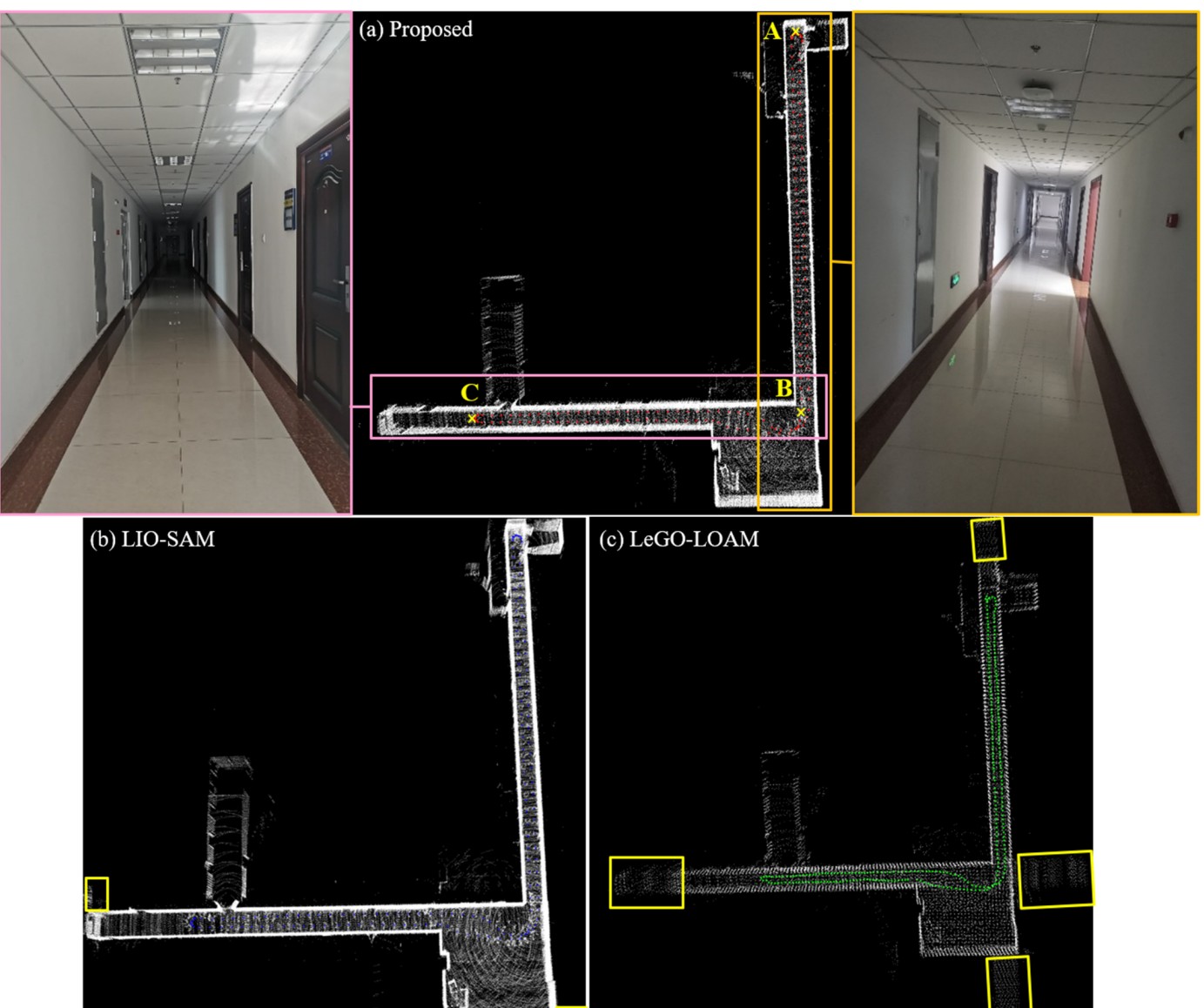

**Figure 6.** Data collection environment, trajectory and mapping results in the indoor corridor. (**a–c**) Trajectories and mappings of the proposed method, the LIO-SAM method and the LeGO-LOAM method, respectively.

3.1.2. Qualitative Analysis with the Underground Coal Mine

Next, experimental data were collected from the underground coal mine, and the length of motion trajectory was about 0.45 km. As shown in Figure 7, there are long and narrow coal mine roadways with different widths, which brings great challenges to the existing LiDAR SLAM methods. Figure 7a–d show the data collection environment of the underground coal mine, and Figure 7e is the motion trajectory (red points) of the mobile robot; the point cloud map of the coal mine was obtained by the proposed method. The statistical analysis showed that the threshold of the degeneration factor in the underground coal mine is 240, and the degeneration state accounts for 45.7% of the total measured data. To quantitatively evaluate the absolute localization accuracy of the proposed method, the coordinate values of 20 reference points (K1-K20, green points) on the motion trajectory were obtained by Total Station. The stopping time interval of the mobile robot was recorded at the corresponding reference point, and then, the average value of the pose estimation result at the reference point was taken as the measurement value of the proposed method. The localization trajectory of the proposed method is close to the reference trajectory points.

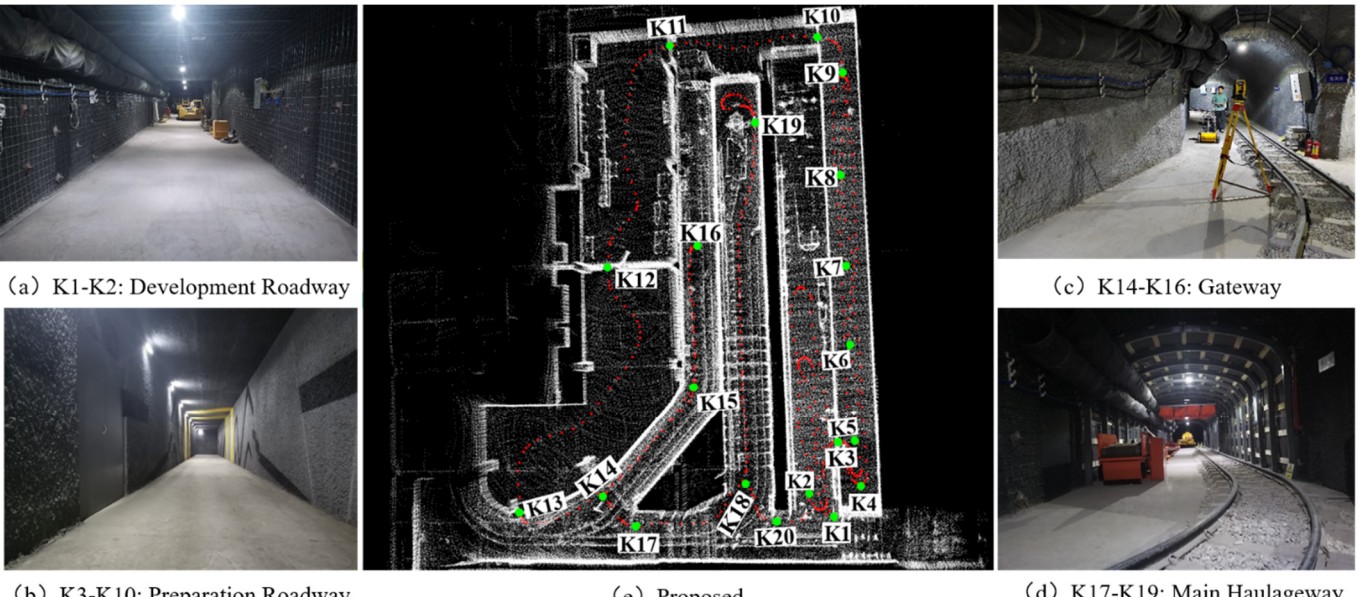

(a) K1-K2: Development Roadway

(b) K3-K10: Preparation Roadway

(e) Proposed

(c) K14-K16: Gateway

(d) K17-K19: Main Haulageway

**Figure 7.** Data collection environment and the mapping result in the underground coal mine. (**a**–**d**) The corresponding data collection environment. (**e**) The mapping result by the proposed method and the layout of reference points.

The localization and mapping results by the proposed method and the LeGO-LOAM and LIO-SAM methods are shown in Figures 8 and 9. The trajectories of LeGO-LOAM and LIO-SAM degenerated significantly in the direction of rotation and translation at the preparation roadway. The degeneration of pitch angle by LeGO-LOAM led to a large trajectory drift and mapping ghosting in the gateway and main haulageway. LIO-SAM mistakenly performed loop closure due to the degeneration of the X-axis direction at the main haulageway, resulting in a large trajectory drift and mapping ghosting of the development roadway and the preparation roadway. In this paper, IMU pre-integration was used to compensate for degeneration, and factor graph optimization was introduced to reduce trajectory drift. The point cloud map constructed by the proposed method can intuitively reflect the actual condition of the roadway environment and had good robustness and accuracy in the underground coal mine. As shown in Figure 8, the trajectory of the proposed method was consistent with the reference trajectory points. LIO-SAM established a wrong loop closure constraint near the start point, which led to the drift of overall trajectory. LeGO-LOAM had a large drift in the Z-axis due to the interference of some similar point clouds and no loop closure constraint.

Figure 10 shows the localization and mapping results by the proposed method, LeGO-LOAM and LIO-SAM in the preparation roadway. Overall, 76.1% of the LiDAR data were in a degenerated state. The localization trajectory of the proposed method is close to the reference trajectory points. It can be seen that the mapping accuracy is high from the thickness of the vertical wall. As shown in Figure 10b, LIO-SAM degenerated in the Y-axis direction, and the localization trajectory has a large drift, resulting in point cloud mismatching and mapping ghosting. LeGO-LOAM degenerated in rotation and translation, and the whole map rotates incorrectly in the yaw direction, resulting in a large trajectory drift and mapping ghosting, as shown in Figure 10c, where there are many wrong walls built around. The proposed method performs degeneration detection and compensation by IMU pre-integration and obtained better localization and mapping results, as shown in Figure 10a.

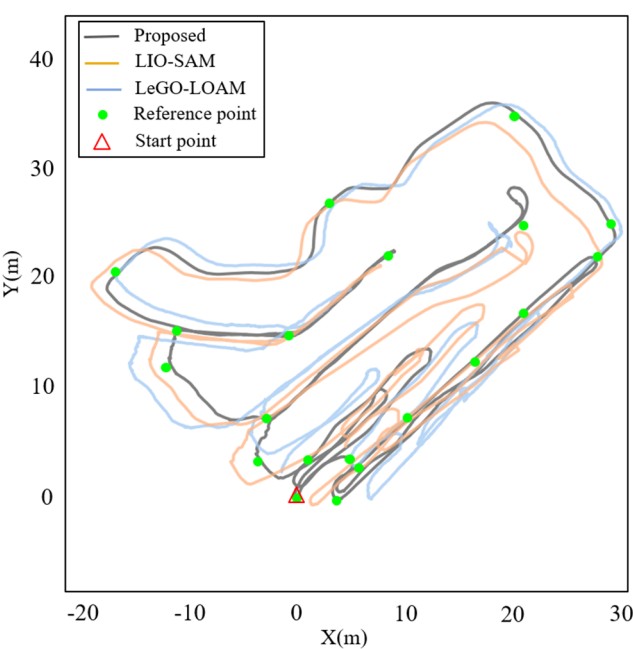

**Figure 8.** Comparison of the trajectories between the proposed method, LeGO-LOAM and LIO-SAM.

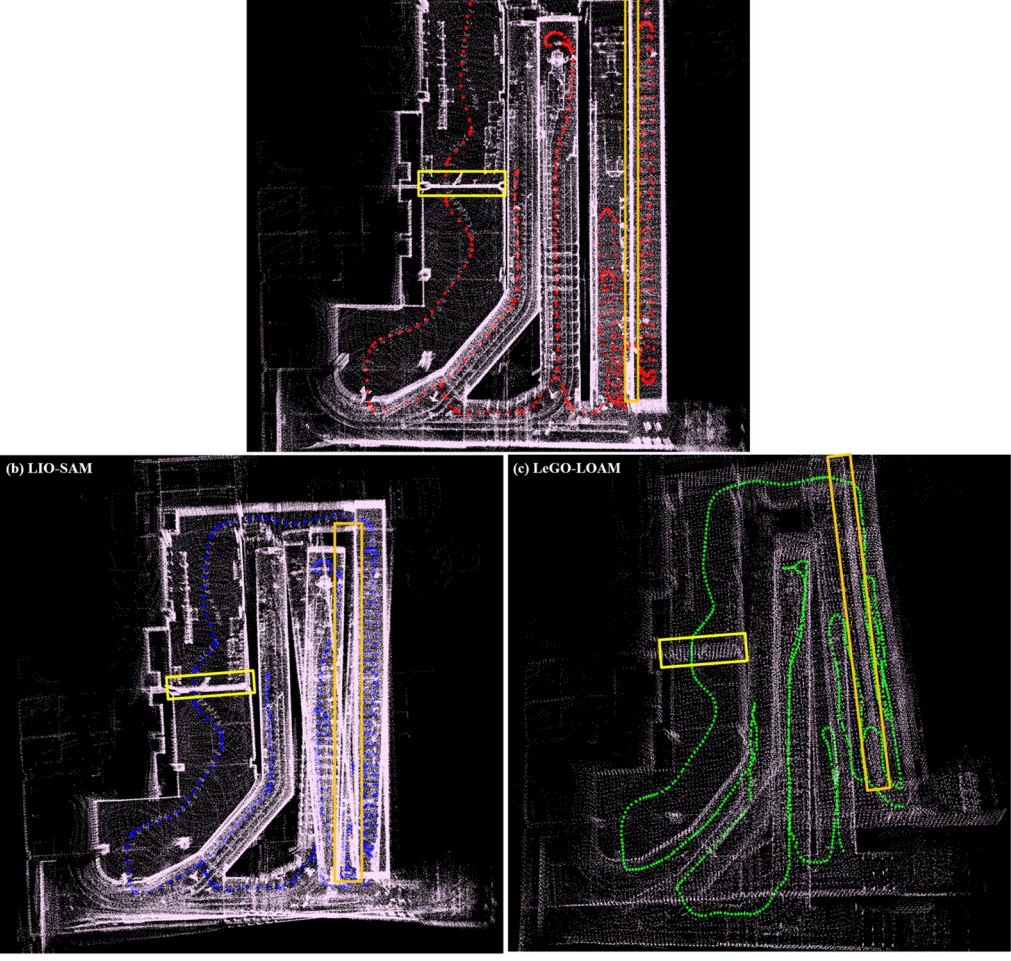

**Figure 9.** Comparison of mapping results between the proposed method and other methods. (**a–c**) Mapping results by the proposed method, LIO-SAM and LeGO-LOAM, respectively.

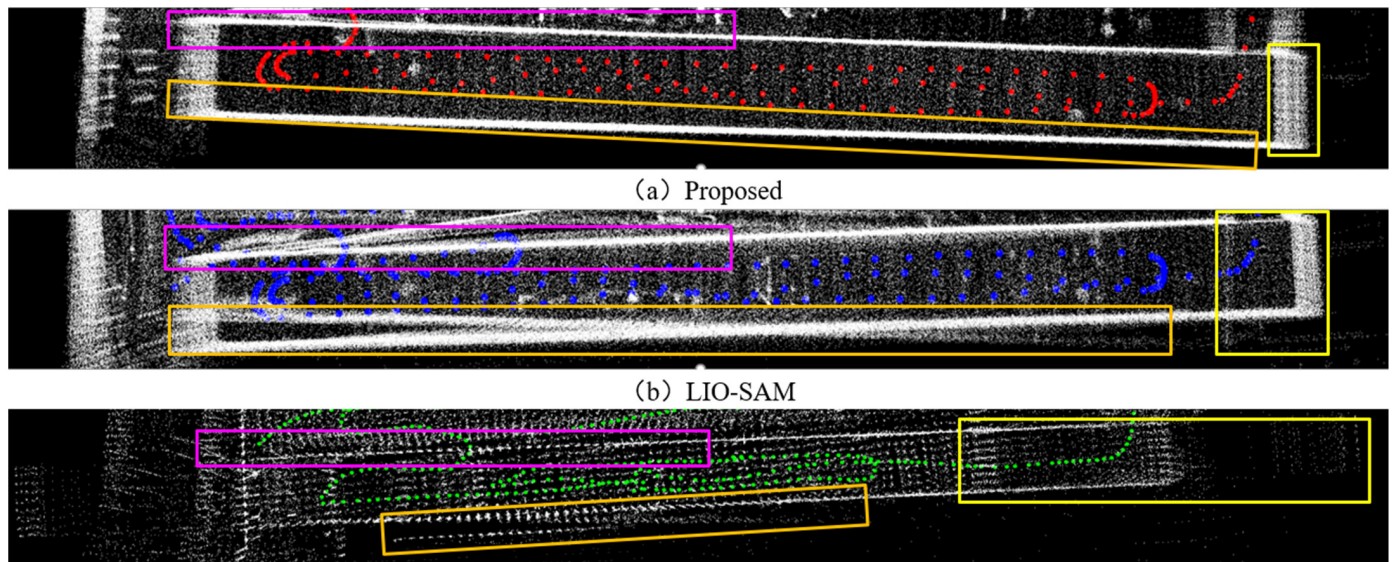

**Figure 10.** Trajectories and mapping results by the proposed method and other methods in the preparation roadway. (**a**–**c**) Trajectories and mapping results by the proposed method, LIO-SAM and LeGO-LOAM, respectively.

*3.2. Quantitative Evaluation*

To quantitatively evaluate the accuracy of the proposed method, a laser range finder was used to measure the distance of AB and BC in the indoor corridor. The mean value measured by the laser range finder was the reference value. The quantitative comparisons of the proposed method, LIO-SAM and LeGO-LOAM are shown in Table 2, where $L_{A\_A}$ represents the distance from the start to the end of the mobile robot. The proposed method showed a higher localization accuracy in the degenerated corridor than the other two methods. The error percentage between the proposed method and the reference value is less than 0.6%, and the distance error of the trajectory closure is 0.07 m. Due to the large windows at the end of the BC segment, the laser passed through the window, resulting in the lack of laser scanning data, which reduced the constraints and made the accuracy of pose estimation in the BC segment slightly lower than that in the AB segment. The sidewalls in the corridor environment only provided lateral and insufficient vertical constraints. Some similar point clouds are prone to mismatching, and the robot cannot determine that it is moving forward. LIO-SAM and LeGO-LOAM were degenerated in the Y-axis direction and had poor localization performance.

**Table 2.** Quantitative comparison of the proposed method, LIO-SAM and LeGO-LOAM (m).

| Length | Reference | LeGO-LOAM | LIO-SAM | Proposed | LeGO-LOAM Percentage | LIO-SAM Percentage | Proposed Percentage |
|--------|-----------|-----------|---------|----------|----------------------|--------------------|--------------------|
| $L_{AB}$ | 38.87 | 34.95 | 38.41 | 38.68 | 10.08% | 1.18% | 0.49% |
| $L_{BC}$ | 36.25 | 32.16 | 35.79 | 36.04 | 11.28% | 1.27% | 0.58% |
| $L_{A\_A}$ | 0.0 | 0.92 | 0.20 | 0.07 | 0.61% | 0.13% | 0.05% |

Figure 11 shows the absolute localization error between the proposed method, LIO-SAM and LeGO-LOAM. The localization accuracy of LeGO-LOAM is poor compared with the proposed method and LIO-SAM, and the maximum translation error in the triaxial direction is more than 1 m. The mean and median of triaxial direction error by the proposed method are lower than those of LIO-SAM. Although the error in the Y-axis direction fluctuates, it is compensated for to a certain extent, and the maximum triaxial direction translation error is smaller than that of LIO-SAM. The localization accuracy of the proposed

method in the X-axis and Z-axis directions is significantly higher than that in LIO-SAM. The mean translation error in the triaxial direction is less than 0.2 m.

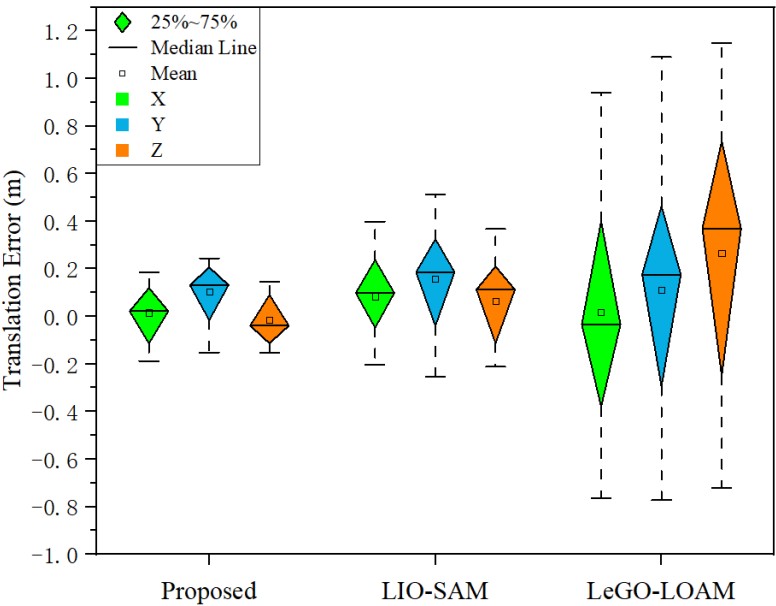

**Figure 11.** Absolute localization errors.

In order to further verify the proposed method, the RMSE was used to evaluate theabsolute accuracy. The results are shown in Table 3. The degeneration of LeGO-LOAM led to the RMSE being more than 0.5 m in the triaxial direction between the measured data and the reference points, and its position RMSE was 0.976 m. The localization accuracy of the proposed method was high, especially in the Z-axis direction, which can reach 0.044 m. The RMSE of position error was 0.161 m. Base on the triaxial direction and position RMSE, the localization accuracy of the proposed method is higher than that of LIO-SAM.

**Table 3.** The RMSE of absolute localization errors(m).

| Method | X | Y | Z | Position |
|---|---|---|---|---|
| LeGO-LOAM | 0.504 | 0.533 | 0.607 | 0.952 |
| LIO-SAM | 0.197 | 0.265 | 0.172 | 0.372 |
| Proposed | 0.084 | 0.130 | 0.044 | 0.161 |

The relative error of the trajectory by the three methods in the three-axis direction is shown in Figure 12. Compared with the proposed method, LeGO-LOAM and LIO-SAM had a certain drift in the X-axis and Y-axis directions, which was caused by the degeneration of the matching process. However, the relative error of the LIO-SAM trajectory was smaller than that of LeGO-LOAM. The main reason is that LIO-SAM fuses IMU information and adds loop closure to suppress cumulative errors. The two methods had great drift in the Z-axis direction, especially for the relative error by LeGO-LOAM, which was as high as 4 m. The mapping of LeGO-LOAM is shown in Figure 9c. There was a large cumulative error in the Z-axis direction after a period of operation. The proposed method detects and compensates for the degenerated scenes in the underground coal mine environment, adds loop closure detection and constructs a global constraint LiDAR SLAM based on factor graph optimization, which obtains a better localization result.

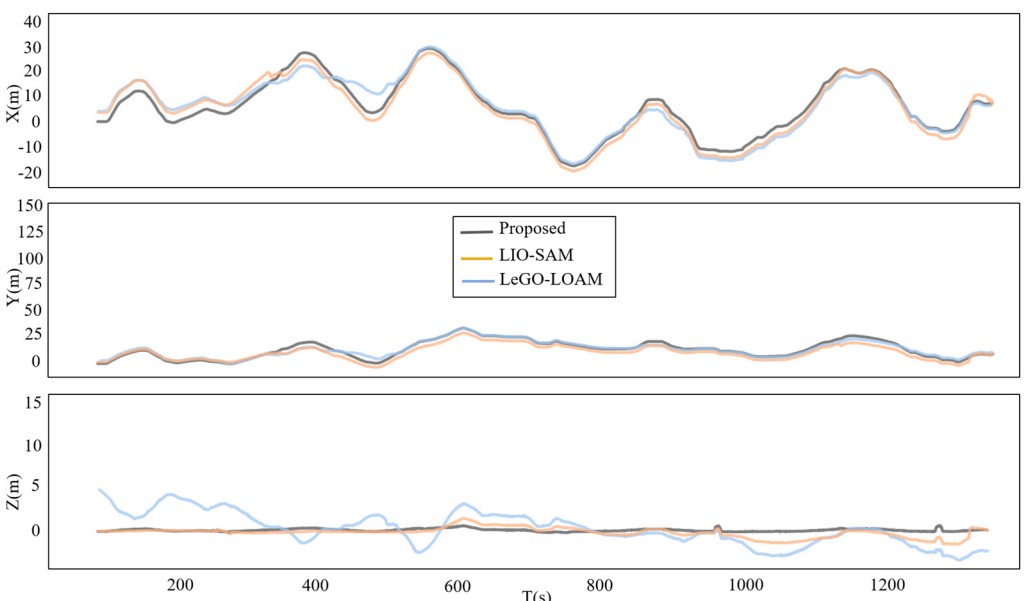

**Figure 12.** The relative error of trajectory in three-axis direction.

### 3.3. Time Performance

The time performance of the proposed method is shown in Figure 13, where P represents the preprocessing, DD represents degeneration detection, DC represents degeneration compensation, FGO represents factor graph optimization and M represents mapping. It can be seen that the preprocessing takes less time, and the degeneration detection consumes little time. The mean and median values of degeneration compensation are no more than 20 ms, and the maximum time for factor graph optimization is no more than 10 ms. The mapping thread takes the most time and can reach 165.75 ms, with an average of 109.97 ms. In the proposed method, the mapping thread receives the point cloud and updates the map at 5 Hz, which can run in real-time.

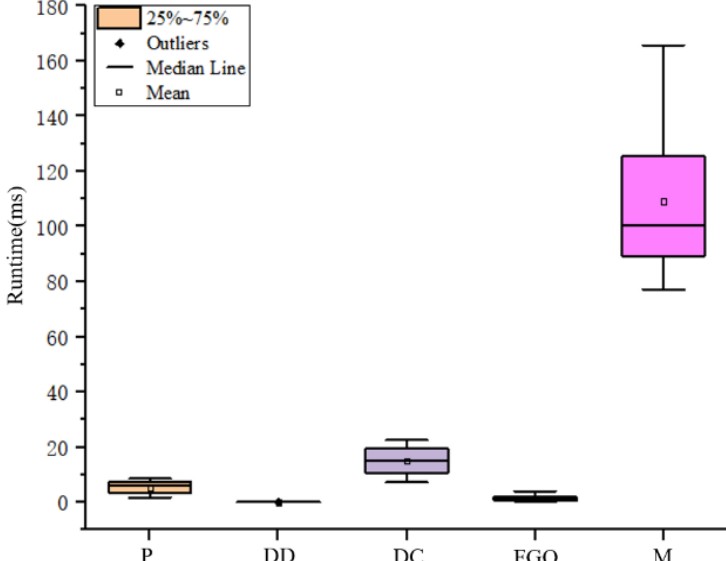

**Figure 13.** Time performance of the proposed method.

### 3.4. Discussion

(1) Pose estimation: To achieve accurate and robust pose estimation, the disturbance model was used to detect the direction and degree of degeneration caused by insufficient line and plane feature constraints for obtaining the factor and vector of degeneration. The

IMU poses were projected onto plane features and fused into new LiDAR poses for local map matching to achieve two-step degeneration compensation. Moreover, loop closure and factor graph optimization were added to suppress the cumulative error of pose estimation. The comprehensive experiments showed that the proposed method was superior to state-of-the-art methods in the underground coal mine. However, the quantitative analysis was limited by the actual situation, as shown in Figure 7e. Only 20 reference points at key positions were selected for quantitative analysis.

(2) Mapping results: The proposed method had a good mapping effect due to the detection and compensation of degeneration states. However, the mapping results by LeGO-LOAM and LIO-SAM in the underground coal mine were relatively inaccurate as the walls were thick and not even aligned, as shown in Figures 9 and 10. The disadvantage here is that it is difficult to make a quantitative comparison. The tight integration of LiDAR and IMU data may further improve SLAM accuracy.

(3) Time performance: It can be seen that the runtime of the proposed method is slightly higher due to the detection and compensation of degeneration, as shown in Figure 13. However, the runtime is still less than 0.1 s, which can run in realtime on low-power embedded systems.

## 4. Conclusions

To address the problem of LiDAR SLAM easily degenerating in the shotcrete surface and symmetrical roadway of underground coal mines, this paper proposed a robust LiDAR SLAM method which detects and compensates for degenerated scenes by integrating LiDAR and IMU data. The disturbance model is used to detect the direction and degree of degeneration for obtaining the factor and vector of degeneration. The pose obtained by IMU pre-integration is projected to plane features for the compensation of rotation state degeneration. The compensated rotation and IMU translation state are fused into a new pose from LiDAR when the translation direction degeneration is detected, which is then used for scan-to-submap matching to achieve two-step degenerated compensation. Lastly, globally consistent LiDAR SLAM is implemented based on factor graph optimization. It can reduce the global cumulated error and improve the trajectory accuracy and map consistency. Extensive experimental results show that the proposed method achieves better robustness than state-of-the-art LiDAR SLAM methods. In the indoor corridor, the error percentage of the proposed method did not exceed 0.6%, and the distance error of trajectory closure was only 0.07 m. In the underground coal mine, the accuracy of pose estimation by the proposed method was the highest in the Z-axis direction, its translation error was only 0.044 m and the absolute position RMSE was 0.161 m. The point cloud map constructed has an excellent performance in integrity and geometric structure authenticity, providing an important reference for underground autonomous navigation and positioning in intelligent mining and safety inspection. In the future, multi-sensor fusion localization and mapping will be further carried out in combination with the degenerated scene in the coal mine to improve the accuracy of localization and mapping.

**Author Contributions:** X.Y. wrote the manuscript. X.Y. designed the experiments and framework. X.L. supervised the entire process of the research and polished the manuscript. W.Y. and H.M. helped organize the paper. J.Z. and B.M. gave constructive advice for the preparation of the paper. All authors have read and agreed to the published version of the manuscript.

**Funding:** This research was funded by the National Natural Science Foundation of China (Grant Nos. 42201484 and 51975468).

**Data Availability Statement:** The data that support the findings of this study are available from the corresponding author upon reasonable request.

**Acknowledgments:** The research in this article was supported by the National Natural Science Foundation of China (Grant Nos. 42201484 and 51975468), which is deeply appreciated.

**Conflicts of Interest:** The authors declare no conflict of interest.

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
