# Peer review of "A Robust LiDAR SLAM Method for Underground Coal Mine Robot with Degenerated Scene Compensation"

_remotesensing, doi:10.3390/rs15010186_

Round 1
Reviewer 1 Report
This article is of interest to readers of this journal and may be accepted. However, I feel that there is a little less evaluation and consideration of accuracy. Also, the notation of the figures and the equations are not clean, so it is difficult to read. It can be accepted with such a minor modification.

Author Response
Response to Reviewer 1 Comments
- Comment: This paragraph is not necessary.
Reply: Thanks very much for your comments. We have removed this paragraph (page 4, lines 164-167).
- Comment: It is difficult to read the characters in the figure, so it is necessary to devise.
Reply: Thanks very much for your comments. We have redesigned Figure 1 (page 5).
- Comment: Blank line.
Reply: Thanks very much for your comments. We have added a blank line (page 5).
- Comment: Overlapping and unreadable.
Reply: Thanks very much for your comments. We have made corrections in the IMU pre-integration section (page 6, lines 212-240).
- Comment: Font size does not match.
Reply: Thanks very much for your comments. We standardized the formula size throughout the text.
- Comment: The items on the x-axis overlap and are difficult to read. Is not the font size all over the place?
Reply: Thanks very much for your comments. We have redesigned Figure 4 (page 10).
- Comment: Are the font types and sizes different?
Reply: Thanks very much for your comments. We have standardized the font type and size throughout the text.
- Comment: Is “,” an alphabetic font?
Reply: Thanks very much for your comments. We have revised “、” to “,” (page 12).
- Comment: Too verbose.
Reply: Thanks very much for your comments. We have removed the redundant words. And checked the full text.
- Comment: Space.
Reply: Thanks very much for your comments. “0.07m” to “0.07 m” (page 20).
- Comment: Lines are thin and hard to read.
Reply: Thanks very much for your comments. We have redesigned Figure 12 (page 23).
- Comment: Can it be made independent as chapter 4 ?
Reply: Thanks very much for your comments. The discussion section is not very much, and it is not suitable to make it a separate chapter in this manuscript (page 24).

Reviewer 2 Report
This paper proposed a robust LiDAR SLAM method, which detects and compensates for the degenerated scenes by integrating LiDAR and Inertial Measurement Unit (IMU) data. The reviewer thinks the topic discussed in this paper is very important, which is of great significance for underground autonomous localization and navigation in intelligent mining and safety inspection. This reviewer sees that a major revision will be needed before being accepted for possible publication. Here are the main comments for the revision.
(1) A key problem in this paper is the lack of introduction to the applicability of the proposed method. Is the application of the method affected by geological conditions? Is this model applicable to all regions?
(2) The introduction section lacks some important references, such as:
Research on Wireless Remote Control for Coal Mine Detection Robot;
Centrifugal model test on a riverine landslide in the Three Gorges Reservoir induced by rainfall and water level fluctuation.
(3) Some Figs are unclear and unexplained. For example, figure 9&10 are not clear. Figure 3 lacks detailed explanation.
(4) The conclusion is not concise and innovative. I believe that the Authors should try to interpret and explain more clearly their results. Some key quantitative conclusions should be supplemented.
(5) In this study, the authors must improve the statements about what is new in their study and what are the contributions to the developments of agriculture.
(6) The text is not clear. The author is suggested to check the professional vocabulary, grammar and spelling of the full text.
Author Response
Response to Reviewer 2 Comments
- Comment: A key problem in this paper is the lack of introduction to the applicability of the proposed method. Is the application of the method affected by geological conditions? Is this model applicable to all regions?
Reply: Thanks very much for your comments. The proposed method can be applied to long straight corridors indoors, underground coal mines with the shotcrete surface and symmetric roadway environment which prone to degeneration. It solves the problem of degeneration when LiDAR point cloud matching in underground coal mines, which can improve the accuracy of localization and mapping (page 4, lines 137-146, 161-163).
- Comment: The introduction section lacks some important references, such as: Research on Wireless Remote Control for Coal Mine Detection Robot; Centrifugal model test on a riverine landslide in the Three Gorges Reservoir induced by rainfall and water level fluctuation.
Reply: Thanks very much for your comments. In the introduction, we added references for localization and mapping of the underground environment by autonomous driving robots (page 3, lines 116-123).
- Comment: Some Figs are unclear and unexplained. For example, figure 9&10 are not clear. Figure 3 lacks detailed explanation.
Reply: Thanks very much for your comments. We have checked all the Figs in the manuscript, and updated the unclear Figs. The underground coal mine roadways are very complex, with four roadways distributed side by side. There are many supporting columns above, and the whole range is large, as shown in the Figure below. The mapping and trajectory of LIO-SAM and LeGO-LOAM have a large drift, and the constructed point cloud map has some overlap, so the point cloud is unclear from Figs. We have redesign the Figure 9 (page 19) and 10 (page 20), and the quality has been improved. Figure 3 (page 8) added the legend and explained the constraints and coordinate system.
The figure shows the side view of the point cloud constructed by the three methods, where the yellow box is the supporting column.
- Comment: The conclusion is not concise and innovative. I believe that the Authors should try to interpret and explain more clearly their results. Some key quantitative conclusions should be supplemented.
Reply: Thanks very much for your comments. We have redesigned and refined the conclusion section. Quantitative conclusions from two environmental experiments were added. In the indoor corridor, the error percentage of the proposed method does not exceed 0.6%, and the distance error of trajectory closure is only 0.07 m. In the underground coal mine, the accuracy of pose estimation by the proposed method is the highest in the Z-axis direction, its translation error is only 0.044 m, and the absolute position RMSE is 0.161 m (page 25, lines 609-617, 619-624).
- Comment: In this study, the authors must improve the statements about what is new in their study and what are the contributions to the developments of agriculture.
Reply: Thanks very much for your comments. We have restated and refined the main contributions in this study (page 3, lines 152-163).
- Comment: The text is not clear. The author is suggested to check the professional vocabulary, grammar and spelling of the full text.
Reply: Thanks very much for your comments. We have carefully checked and revised the professional vocabulary, grammar and spelling of the full text. Moreover, we have sought professional language polishing and retouching services.

Reviewer 3 Report
Dear Editor, Dear Authors,
The manuscript by Yang et al. presents a method for coupling LiDAR odometry with inertial measurement, to:
a) correct for the LiDAR point cloud distortion caused by the robot movement,
b) add an additional estimate of short-term pose change based on IMU preintegration as a constrain in the factor graph,
c) detect the degeneration of the scene in the point cloud data and correct the LiDAR pose in such scenes.
The research topic tackled in the paper is important and the original method proposed by the Authors might be of high interest to the readers, with the potential for multiple practical applications. However, I see several key issues in the methodology and the results that should be addressed before accepting the manuscript for publication and some other, minor ones.
My main concerns are:
1. The literature review, while properly describing and containing important, relevant publications, focuses strongly on feature-based methods, which match linear and planar features in the point cloud. In the comparison with the proposed method, only this family of methods was investigated. However, especially for low-cost lidars such as 16-line scanners, methods based on GICP and NDT generally show good results; it would be appropriate to investigate and include at least in the review methods such as [1], [2], elaborating why they were not chosen for the comparison in the further analysis.
2. Authors of [3] proposed a different, but similar solution of fusing LiDAR with IMU to successfully perform SLAM also in degenerated scenes. Due to the similarity in the problem being solved, it would be important to compare their approach with this presented by the Authors: what is the advantage of the proposed method over [3]? Which approach could be more universal? Etc.
3. The motivation behind the step presented in lines 213-220 is somewhat confusing. The Authors split estimation of pose parameters in roll/pitch/tz (which, generally, for a UGV should be less variable than the other 3) and yaw/tx/ty, but do not provide sufficient reasoning for utilizing only linear/planar features for estimating each of the parameter groups. As this step seems to be unique among the approaches present in the literature, it should be backed by some more proof to elevate the scientific soundness and uniqueness of the article.
4. I see 2 problems in the degeneration compensation algorithm. Firstly, the threshold value for the degeneration factor is roughly assumed to be 180 based on the plot of degeneration factor values distribution in the histogram. As I understood from the manuscript, this value was not calculated using any particular clustering method, based only on a degeneration factor distribution in the indoor tests. Later on, it is not shown how the distribution looks in the real underground environment, thus not providing evidence, that such clustering is also visible in the robot's operational conditions. The Authors may also elaborate if such boundary point selection could be automated with any existing clustering algorithms, which would increase the universality of the system (e.g. in future works).
5. The degeneration algorithm (Algorithm 1) is not clear to me: the conditions in the nested while loops are identical, without any indication of Dt changing in lines 4-6 or 8-11.
6. As we do not know the ground truth (e.g. a metric floor plan) of the results shown in Figure 6, does the proposed method show a higher angular error of matching corridors AB and BC than LeGO-LOAM? The local point cloud coherence is clearly better, but this aspect is not clearly visible and not described.
7. Are the trajectories presented in Figure 8 starting from the same position and heading? The plots in Figure 12 suggest otherwise, as only the trajectory obtained with the method proposed by the Authors starts from point (0,0,0). If not, this should be corrected, as it introduces an additional, artificial error to other trajectories.
8. The Authors describe the motivation and applicability of their work strictly in the context of underground coal mining. However, the presented problem of degenerated scenes is not exclusive to such conditions and widely encountered in almost any other type of underground mine and other industrial indoor facilities (e.g. power plants). If successful, I think the presented approach has much broader application potential than described by the Authors, so I suggest generalizing the title of the work and description of the method design in terms of the robot's working conditions.
My other minor suggestions are:
a) the grammar of the manuscript could be improved; even in the first sentence of the introduction, there is a subject-verb disagreement (“…fields…” – ‘”…is…”). Please check the overall grammatical correctness of the paper.
b) Please also check for typos, e.g. Figure 4: “Precentage”.
c) A legend for arrows and boxes in Figure 3 could be provided.
d) Shouldn’t percentage errors in Table 2 be also reported for L(A_A), as the absolute error divided by the total path length (ABCBA)?
e) The font seems to be inconsistent for some symbols in Algorithm 1.
f) For all visualizations of the point cloud results, the legend/description of boxes could be provided. Are those point clouds rendered with the same settings? Would it be possible to provide a higher resolution of e.g. Figure 9?
g) There is no statement of data availability in the manuscript. Also, if the implementation of the proposed solution is based on or an extension of another already published work, it should be clearly indicated in the paper.
To sum up, I believe that the work of the Authors could be of high interest to Remote Sensing readers. However, the review, methodology description and result presentation are lacking some important elements, thus I recommend reconsidering the acceptance of the manuscript by Yang et al. after major revision.
[1] Koide, Kenji & Yokozuka, Masashi & Oishi, Shuji & Banno, Atsuhiko. (2022). Globally Consistent and Tightly Coupled 3D LiDAR Inertial Mapping. 10.1109/ICRA46639.2022.9812385.
[2] M. Li, H. Zhu, S. You, L. Wang and C. Tang. (2019). Efficient Laser-Based 3D SLAM for Coal Mine Rescue Robots. IEEE Access, vol. 7, pp. 14124-14138, doi: 10.1109/ACCESS.2018.2889304.
[3] W. Xu and F. Zhang. (2021). FAST-LIO: A Fast, Robust LiDAR-Inertial Odometry Package by Tightly-Coupled Iterated Kalman Filter. IEEE Robotics and Automation Letters, vol. 6, no. 2, pp. 3317-3324, April 2021, doi: 10.1109/LRA.2021.3064227.
Author Response
Response to Reviewer 3 Comments
- Comment: The literature review, while properly describing and containing important, relevant publications, focuses strongly on feature-based methods, which match linear and planar features in the point cloud. In the comparison with the proposed method, only this family of methods was investigated. However, especially for low-cost lidars such as 16-line scanners, methods based on GICP and NDT generally show good results; it would be appropriate to investigate and include at least in the review methods such as [1], [2], elaborating why they were not chosen for the comparison in the further analysis.
Reply: Thanks very much for your comments. Just as you said, NDT or GICP methods can get good matching results. However, in the experiments of this study we found that it is more efficient to use feature-based matching method in the front-end odometer part. Therefore, we consider feature extraction and matching of LiDAR point clouds, which can meet the accuracy requirements and run in real-time. In the back-end optimization we use the GICP algorithm for the scan-to-submap matching to obtain higher localization and mapping accuracy (page 13, lines 395-396). Supplementary literature and refinements have been made in the introduction section (page 2, lines 92-94).
- Comment: Authors of [3] proposed a different, but similar solution of fusing LiDAR with IMU to successfully perform SLAM also in degenerated scenes. Due to the similarity in the problem being solved, it would be important to compare their approach with this presented by the Authors: what is the advantage of the proposed method over [3]? Which approach could be more universal? Etc.
Reply: Thanks very much for your comments. Authors of [3] proposed a computationally efficient and Robust LiDAR-inertial odometry framework using a tightly-coupled iterated extended Kalman filter to allow robust navigation in fast-motion, noisy or cluttered environments where degeneration occurs. In contrast, this study proposed a robust LiDAR SLAM method, which detects and compensates for the degenerated scenes by integrating LiDAR and IMU data. It can be used in the more complex underground coal mine environment. We have compared and refined the overview in the introduction section (page 2, lines 95-98).
- Comment: The motivation behind the step presented in lines 213-220 is somewhat confusing. The Authors split estimation of pose parameters in roll/pitch/tz(which, generally, for a UGV should be less variable than the other 3) and yaw/tx/ty, but do not provide sufficient reasoning for utilizing only linear/planar features for estimating each of the parameter groups. As this step seems to be unique among the approaches present in the literature, it should be backed by some more proof to elevate the scientific soundness and uniqueness of the article.
Reply: Thanks very much for your comments. The underground coal mine roadway has more plane feature points than line feature points in our experiments. In this study, we used a two-step optimization for pose estimation. First, the plane feature matching was used to determine the and derived a more accurate pose. Then the was used as a constraint for line feature matching to achieve overall optimization and improve the accuracy of pose estimation (page 9, lines 292-301).
- Comment: I see 2 problems in the degeneration compensation algorithm. Firstly, the threshold value for the degeneration factor is roughly assumed to be 180 based on the plot of degeneration factor values distribution in the histogram. As I understood from the manuscript, this value was not calculated using any particular clustering method, based only on a degeneration factor distribution in the indoor tests. Later on, it is not shown how the distribution looks in the real underground environment, thus not providing evidence, that such clustering is also visible in the robot's operational conditions. The Authors may also elaborate if such boundary point selection could be automated with any existing clustering algorithms, which would increase the universality of the system (e.g. in future works).
Reply: Thanks very much for your valuable comments. in Fig.4 is the empirical value statistically analyzed by multiple experiments in the indoor corridor. The proposed method in this manuscript is able to estimate the pose of a coal mine underground robot in a degeneration state, it can also estimate the pose of the robot when no degeneration occurs. The statistics present a clustering effect and we take the cut-off clustering value as a threshold (page 10, lines 321-325). The same method was used to determine the degeneration threshold in the underground coal mine environment. We will consider automating the boundary point selection to increase the system's universality in future work.
- Comment: The degeneration algorithm (Algorithm 1) is not clear to me: the conditions in the nested while loops are identical, without any indication of Dt changing in lines 4-6 or 8-11.
Reply: Thanks very much for your valuable comments. We are sorry for the misunderstanding. We have revisited the logic of Algorithm 1 (page 12). We changed the “ while ” to “ if ” and replaced the loop statement with a judgment statement. The short term IMU poses were used to fuse and compensate for the degeneration regardless of which direction the degeneration occurred.
- Comment: As we do not know the ground truth (e.g. a metric floor plan) of the results shown in Figure 6, does the proposed method show a higher angular error of matching corridors AB and BC than LeGO-LOAM? The local point cloud coherence is clearly better, but this aspect is not clearly visible and not described.
Reply: Thanks very much for your valuable comments. The ground truth of the mobile robot in the corridor cannot be obtained accurately. Therefore, we indirectly reflect the accuracy of the proposed method in this manuscript by the overall effect of the mapping results and the point cloud thickness of local walls in the indoor corridor. Moreover, we have compared the proposed with the typical methods, such as LIO-SAM and LeGO-LOAM. As shown in Figure 6, LIO-SAM drifts in the X-axis direction of AB and BC segments, and there is also a ghosting in the map. A significant degeneration occurred in LeGO-LOAM, especially in the Y-axis direction of the AB and BC segments, and the trajectory is shortened in the forward direction of the robot (page 14, lines 421-435).
- Comment: Are the trajectories presented in Figure 8 starting from the same position and heading? The plots in Figure 12 suggest otherwise, as only the trajectory obtained with the method proposed by the Authors starts from point (0,0,0). If not, this should be corrected, as it introduces an additional, artificial error to other trajectories.
Reply: Thanks very much for your valuable comments. Trajectories generated by the proposed method, LIO-SAM and LeGO-LOAM are started from the same position and heading. We have made relevant revisions, as shown in Figure 8 (page 17). LIO-SAM has an incorrect loop closure, and the trajectory drifts. LeGO-LOAM has no loop closure, which produces incorrect matching and the trajectory cannot be closed. Therefore, the trajectories of LIO-SAM and LeGO-LOAM in the figure do not appear to be aligned with the proposed method. In fact, they are generated by different methods from the same position and heading with the same setting parameters. We have made relevant revisions to the relative error of trajectories in Figure 12 (page 23).
- Comment: The Authors describe the motivation and applicability of their work strictly in the context of underground coal mining. However, the presented problem of degenerated scenes is not exclusive to such conditions and widely encountered in almost any other type of underground mine and other industrial indoor facilities (e.g. power plants). If successful, I think the presented approach has much broader application potential than described by the Authors, so I suggest generalizing the title of the work and description of the method design in terms of the robot's working conditions.
Reply: Thanks very much for your valuable comments. Just as you suggested, degeneration may exist in almost any other type of underground mine and other industrial indoor facilities, and we only conducted experiments in long indoor corridors and underground coal mines. In the future, we will make more attempts to further extend and improve our method to make it has a much broader application (page 2, lines 56-66, 137-146 ).
- Comment: the grammar of the manuscript could be improved; even in the first sentence of the introduction, there is a subject-verb disagreement (“…fields…” – ‘”…is…”). Please check the overall grammatical correctness of the paper.
Reply: Thanks very much for your comments. We have carefully checked and revised the grammar, professional vocabulary and spelling of the full text. Moreover, we have sought professional language polishing and retouching services.
- Comment: Please also check for typos, e.g. Figure 4: “Precentage”.
Reply: Thanks very much for your comments. We have carefully checked and revised the spelling of the full text. For example, we have revised “Precentage” to “Percentage” in Figure 4 (page 10).
- Comment: A legend for arrows and boxes in Figure 3 could be provided.
Reply: Thanks very much for your comments. We have revised and added a legend for Figure 3 (page 8).
- Comment: Shouldn’t percentage errors in Table 2 be also reported for L(A_A), as the absolute error divided by the total path length (ABCBA)?
Reply: Thanks very much for your valuable comments. We have revised and supplemented the absolute error percentage of in Table 2 (page 21).
- Comment: The font seems to be inconsistent for some symbols in Algorithm 1.
Reply: Thanks very much for your valuable comments. We have revised the font in algorithm 1 (page 12).
- Comment: For all visualizations of the point cloud results, the legend/description of boxes could be provided. Are those point clouds rendered with the same settings? Would it be possible to provide a higher resolution of e.g. Figure 9?
Reply: Thanks very much for your valuable comments. We have revised the description of boxes. All point cloud maps and trajectories in this study are rendered with the same settings. And we have recreated Figure 9 to make it has a higher resolution (page 19).
- Comment: There is no statement of data availability in the manuscript. Also, if the implementation of the proposed solution is based on or an extension of another already published work, it should be clearly indicated in the paper.
Reply: Thanks very much for your valuable comments. We added a statement of data availability after the conclusion. The data that support the findings of this study are available from the corresponding author upon reasonable request. And the implementation of the proposed solution was done independently by our team.

Reviewer 4 Report
In this paper, a SLAM method based on LiDAR data is proposed for underground coal mines. The work mainly solve the problem of the degeneration. The method gets good performance, but there are some revisions that need to be made.
(1) In last paragraph of the introduction, the authors did not clarify the problem of the existing methods. It is hard to see the difference between the proposed method and the existing methods.
(2) The process of the feature matching is not described in the paper.
(3) In the experiments, the reference numbers of the LeGO-LOAM and LIO-SAM methods are not given.
(4) The experimental section lacks enough explaination. The authors need to give the reasons why the other methods get poor performance.
(5) For the time performance, the comparison with other methods is not presented.
(6) In table 2, the results of the LIO-SAM method are not listed.
(7) Some sentences are incorrect, such as line 469.
Author Response
Response to Reviewer 4 Comments
- Comment: In last paragraph of the introduction, the authors did not clarify the problem of the existing methods. It is hard to see the difference between the proposed method and the existing methods.
Reply: Thanks very much for your valuable comments. We have revised and refined the last paragraph of the introduction. Existing methods are prone to degeneration on shotcrete surfaces and symmetrical roadways. And the proposed method in this manuscript is used to solve this problem. LiDAR rotation state degeneration is compensated by projecting IMU poses onto plane features. When the degeneration is detected in the translation direction, the compensated rotation state and IMU translation state are fused into a new LiDAR pose, which is then used for scan-to-submap matching to achieve two-step degeneration compensation (page 3, lines 137-163).
- Comment: The process of the feature matching is not described in the paper.
Reply: Thanks very much for your valuable comments. We have added the process of feature matching in Section 2.4. We establish constraints on the minimum point-to-line and point-to-plane distances, respectively. The pose is calculated by nonlinear iterative objective function (page 9, lines 280-291).
- Comment: In the experiments, the reference numbers of the LeGO-LOAM and LIO-SAM methods are not given.
Reply: Thanks very much for your valuable comments. We are sorry for the misunderstanding. We have given the reference numbers of the LeGO-LOAM [14] and LIO-SAM [21] methods in the introduction (page 2). Therefore, in the experimental part we did not give the reference numbers.
- Comment: The experimental section lacks enough explaination. The authors need to give the reasons why the other methods get poor performance.
Reply: Thanks very much for your valuable comments. In the degenerated scene, the width of both sides of the indoor corridor is equal, which easily leads to failure of point cloud matching. The robot mistakenly believed that it did not move forward, then obtained a map with a smaller distance than the actual roadway (page 14, lines 422-435, 521-525). In the underground coal mine environment, LIO-SAM established a wrong loop closure constraint near the start point, which led to the drift of overall trajectory. LeGO-LOAM had a large drift in the Z-axis due to the interference of some similar point clouds and no loop closure constraint (page 16, lines 477-480).
- Comment: For the time performance, the comparison with other methods is not presented.
Reply: Thanks very much for your valuable comments. The time performance of each part of the proposed method after adding degeneration detection and compensation is counted. In this manuscript, the proposed method can meet the requirements of real-time operation and achieve accurate and reliable SLAM in underground coal mines. Therefore, we did not compare with other methods in terms of time performance (page 23, lines 571-576). In the future, we will perform a detailed analysis of time performance with other methods.
- Comment: In table 2, the results of the LIO-SAM method are not listed.
Reply: Thanks very much for your valuable comments. We have supplemented and revised the experiments with the LIO-SAM method and listed its experimental results in Table 2 (page 20).
- Comment: Some sentences are incorrect, such as line 469.
Reply: Thanks very much for your valuable comments. We have carefully checked and revised the grammar, professional vocabulary and spelling of the full text. Moreover, we have sought professional language polishing and retouching services.

Round 2
Reviewer 2 Report
This paper has been revised as required, which can be accepted now.
Reviewer 3 Report
I believe the Authors have made significant changes to the manuscript, improving its overall scientific soundness and readability.
I recommend the acceptance of their work for publication.
Reviewer 4 Report
The authors have made revision to their manuscript. I think it could be accepted for publication.